# Water binding and hygroscopicity in π-conjugated polyelectrolytes

Cindy Guanyu Tang[1], Mazlan Nur Syafiqah[2], Qi-Mian Koh[2], Mervin Chun-Yi Ang [2], Kim-Kian Choo[2], Ming-Ming Sun[2], Martin Callsen[1], Yuan-Ping Feng [1], Lay-Lay Chua [2] ✉, Rui-Qi Png[1] ✉ & Peter K. H. Ho [1] ✉

The presence of water strongly influences structure, dynamics and properties of ion-containing soft matter. Yet, the hydration of such matter is not well understood. Here, we show through a large study of monovalent π-conjugated polyelectrolytes that their reversible hydration, up to several water molecules per ion pair, occurs chiefly at the interface between the ion clusters and the hydrophobic matrix without disrupting ion packing. This establishes the appropriate model to be surface hydration, not the often-assumed internal hydration of the ion clusters. Through detailed analysis of desorption energies and O–H vibrational frequencies, together with OPLS4 and DFT calculations, we have elucidated key binding motifs of the sorbed water. Type-I water, which desorbs below 50 °C, corresponds to hydrogen-bonded water clusters constituting secondary hydration. Type-II water, which typically desorbs over 50–150 °C, corresponds to water bound to the anion under the influence of a proximal cation, or to a cation–anion pair, at the cluster surface. This constitutes primary hydration. Type-III water, which irreversibly desorbs beyond 150 °C, corresponds to water kinetically trapped between ions. Its amount varies strongly with processing and heat treatment. As a consequence, hygroscopicity—which is the water sorption capacity per ion pair—depends not only on the ions, but also their cluster morphology.

Sorbed moisture is ubiquitously present in ion-containing soft matter. Polyelectrolytes and ionomers—which both comprise polymer backbones tethered with fixed ion groups counterbalanced by free ions—typically show an ambient molar water sorption capacity of the order of their ion density[1,2]. Here, fixed ions refer to those that cannot be exchanged from the polymer, unlike free ions. Polymer electrolytes, which are polymers infused with salts, also show similarly large amounts of sorbed moisture[3–5]. Amongst these materials, π-conjugated polyelectrolytes have attracted the most interest[6–11]. Their highly tailorable semiconducting polymer backbones open numerous applications in bioelectronics[12], batteries[13,14], and semiconductor devices[15,16]. However, sorbed moisture can complicate

these applications. It has been implicated in photo-doping[17], *operando* interfacial doping/dedoping[18–20], and charge-carrier trapping[21–23]. Thus, it is crucial to understand the hydration of these materials.

At first sight, the hydration of conventional polyelectrolytes appears well understood. Their constituent ions assemble into hydrophilic clusters inside a hydrophobic matrix[24,25], which is the case also for conjugated polyelectrolytes[26,27]. The 'standard' view is that the first sorbed water molecules bind to ions in the interior of these ion clusters, especially the cations, due to their large negative enthalpy of hydration, to give solvent-shared ion pairs. This model, first popularized by Zundel and co-workers, may be denoted 'bulk

[1]Department of Physics, National University of Singapore, Lower Kent Ridge Road, S117550 Singapore, Singapore. [2]Department of Chemistry, National University of Singapore, Lower Kent Ridge Road, S117552 Singapore, Singapore. ✉e-mail: chmcll@nus.edu.sg; ruiqi@nus.edu.sg; phyhop@nus.edu.sg

hydration' or 'internal hydration'[28,29]. The subsequently sorbed water molecules progressively generate solvent-separated ion pairs, and then a concentrated ionic solution[30]. This final stage is not in doubt for water-soluble polyelectrolytes, and for strongly water-swellable perfluorosulfonate ionomers, such as Nafion and Nafion-M[31–34].

The degree of hydration can be expressed by the hydration number $n_w$, which is the mole ratio of water molecules to ion formula units. In the case of monovalent polyelectrolytes, which have singly-charged cations balanced by single-charged anions, this is given by the mole ratio of water molecules to ion pairs. In the Zundel model, internal hydration is taken to occur from the onset of hydration, that is from the lowest $n_w$ values. Pioneering infrared studies in the 1960s–70 s found that the frequency of the water O–H stretching vibration ($\nu$OH) and $n_w$ depend on both the cation and anion[28,29]. These features suggest a strong interaction between sorbed water and both ions from the onset of hydration. Subsequent thermal studies in the 1980s–90s apparently supported this interpretation[35–37]. The first sorbed water molecules are non-freezable, which suggests 'bound water' tightly held between the ions. Only when $n_w$ exceeds a threshold, which is about seven for poly(styrenesulfonate sodium) (PSSNa) and Nafion-Na, does the 'excess' water molecules become freezable or 'free'.

Yet, a number of puzzling features have been uncovered recently. Neutron reflectometry of polyelectrolyte multilayer films found the first sorbed water molecules to fill voids to give so-called "void water", while the latter sorbed molecules give "swelling water"[38,39]. Unless the interior of the ion clusters comprises significant voids, this would appear incompatible with internal hydration. Dielectric relaxation spectroscopy of Nafion-Na found the first sorbed water molecules to be more not less orientationally mobile than the latter sorbed molecules[40]. Again, this feature is incongruous with the internal hydration model.

Nevertheless, internal hydration clearly occurs in ionic liquids[41–43]. At a low $n_w \lesssim 0.2$, water is dispersed throughout the ionic liquid as solitary molecules bridging adjacent anions by hydrogen bonding. For hydrophobic ionic liquids, such as 1-hexyl-3-methylimidazolium bis(-trifluoromethanesulfonyl)imide, hydration is limited to $n_w \lesssim 0.3$ even at saturation with water[44–46]. For hydrophilic ionic liquids, on the other hand, such as 1-butyl-3-methylimidazolium perchlorate, $n_w$ can be as large as several water molecules per ion pair for a relative humidity (RH) of 65–70% at room temperature, giving solvent-shared and solvent-separated ion pairs[47–51].

Whether monovalent polyelectrolytes hydrated up to the moderately hydrated regime of several water molecules per ion pair also show similar interior hydration is unclear. This question is scientifically and technologically important, because it relates to hygroscopicity control and hydration structure of the material in ambient air (*vide infra*). We became interested in this question because of its relevance to several phenomena we were studying, including small ion clusters acting as universal electron donors[16], stability of ultrahigh-workfunction hole-doped polymers[15,52], and efficiency of the nitrene photocrosslinking mechanism[53–55].

In this work, we report an extensive study of monovalent conjugated polyelectrolytes leading to the establishment of the surface hydration model as the dominant one for monovalent polyelectrolytes up to the moderately hydrated regime. We focussed on monovalent systems because of their chemical simplicity and technological relevance. We surveyed ions across the Hofmeister series[42,56,57]: cations in the set of {Li$^+$, Na$^+$, K$^+$, Cs$^+$, TMA$^+$, TEA$^+$, TPP$^+$; –NMe$_3^+$}, where TMA$^+$ is tetramethylammonium, TEA$^+$ is tetraethylammonium, TPP$^+$ is tetraphenylphosphonium, and –NMe$_3^+$ is tethered trimethylammonium; and anions in the set of {TfO$^-$, TFSI$^-$, BArF$^-$; –CH$_2$SO$_3^-$, –CH$_2$SO$_2$N$^-$SO$_2$CF$_3$, –CH$_2$SO$_2$N$^-$SO$_2$CF$_2$CF$_3$}, where TfO$^-$ is triflate (CF$_3$SO$_3^-$); TFSI$^-$ is bis(trifluoromethanesulfonyl)imide ((CF$_3$SO$_2$)$_2$N$^-$); BArF$^-$ is B(C$_6$H$_3$(*m*-CF$_3$)$_2$)$_4^-$, and –CH$_2$SO$_2$N$^-$SO$_2$R$_F$, where R$_F$ ∈ {CF$_3$,

CF$_2$CF$_3$}, is tethered perfluoroalkylsulfonylimidosulfonyl (denoted 'R$_F$SIS' here). Most of these polyelectrolytes were synthesized in our laboratories. We shall first discuss the experimental evaluation of hygroscopicity and water desorption energies, then theoretical insights gained from OPLS4 molecular force-field simulations and DFT quantum chemical calculations, and finally the synthesis of all results to establish new fundamental insights into the initial hydration structure of monovalent polyelectrolytes.

## Results and discussion
### Thermogravimetry (TG)

TG is the simplest technique to probe water desorption. But it suffers from poor resolution. For example, a set of thermograms for mTFF-SO$_3$-Na powder, a key model of sulfonate conjugated polyelectrolytes, is shown in Fig. 1a. The chemical structure of this polymer is shown in S/N 3, Supplementary Table 1, and computed ion cluster morphology in Supplementary Fig. 1, which resembles those of other conjugated polyelectrolytes[26,27]. The sample was scanned from room temperature to 150 °C in flowing nitrogen, at a temperature ramp (d$T$/d$t$) of between 1 and 10 °C min$^{-1}$, cooled to room temperature (22 °C) in nitrogen, exposed to ambient air for 1 h (RH, 65–70%), re-scanned at a different d$T$/d$t$, and the cycle repeated a few more times. All the thermograms show that water—confirmed by tandem mass spectrometry—desorbs continually over this temperature range, without splitting into distinct steps. However, the derivative thermograms, i.e., −d$m$/d$T$ vs $T$, where $m$ is normalized mass, clearly reveal two regimes, one with divergent behaviour that occurs below 60 °C, and the other with convergent behaviour above this temperature (Fig. 1b). This suggests two fractions of water are involved. The fraction that desorbs over the higher temperature range exhibits a d$m$/d$T$ characteristic determined by temperature alone, while that at the lower range a more complex behaviour. Nevertheless, similar mass loss was recorded in each run except the first, indicating that hygroscopicity is well-behaved, except the first run when the sample was not fully equilibrated in the ambient.

To resolve these two regimes, we devised a simple 'segmented TG' protocol. An isothermal segment is applied to desorb the lower water fraction, then a temperature ramp is applied to desorb the higher fraction(s) in sequence, and a final isothermal segment to fully desorb the highest fraction at a temperature below the polymer degradation temperature. The following conditions provide a good balance between resolution power and time:

a. jump to initial temperature $T_i$ of 35 °C and hold for 10 min;
b. heat at 5 or 10 °C min$^{-1}$ to a final temperature $T_f$ (230 °C or lower) and hold for 10 min.

The resulting segmented thermogram for the mTFF-SO$_3$-Na powder now exhibits three water loss steps (Fig. 1c). The first one is characterized by $n_w$ of 1.2 H$_2$O per ion pair; second one, 1.0 H$_2$O/ion pair, with extrapolated onset and end temperatures of 45 °C and 135 °C, respectively; third one, 0.6 H$_2$O/ion pair, with extrapolated onset of 180 °C. These data are compiled at S/N 3 in Supplementary Table 1, together with sample preparation notes. After the first scan, the sample was cooled in flowing nitrogen, equilibrated in the ambient and re-scanned. The second thermogram turns out similar with $n_w$ of 0.9 and 1.0 H$_2$O/ion pair for the first two steps, respectively, but $n_w < 0.02$ for the third step. This indicates irreversible elimination of the binding sites for the highest water fraction.

This three-step desorption behaviour is also found in other polyelectrolytes. For example, when the hydrophilic sulfonate moiety of the polymer is substituted by the hydrophobic pentafluoroethylsulfonylimidosulfonyl moiety, the steps are similar albeit smaller in size (Fig. 1d; see also Supplementary Table 1: S/N 18).

In this way, we have studied 29 conjugated polyelectrolytes, including charge-doped forms of some of them, together with two non-conjugated polyelectrolytes as references (S/N 1 PSSNa, and S/N2

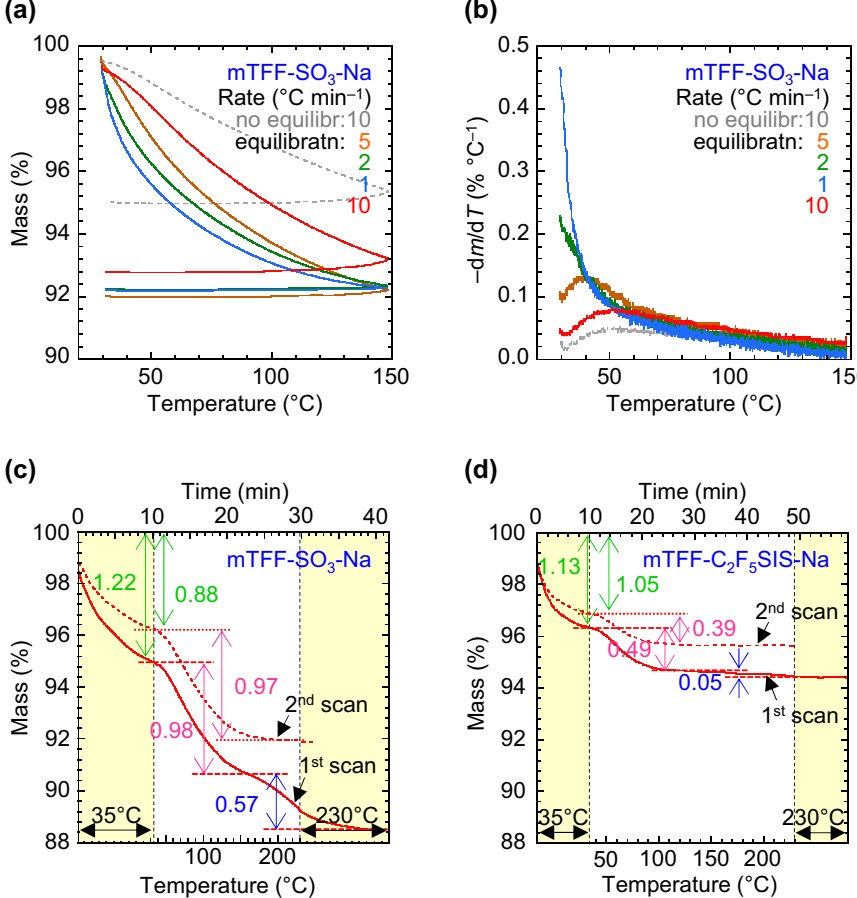

**Fig. 1 | Water desorption thermogravimetry (TG) analysis. a** TG of mTFF-SO$_3$-Na powder: first scan at 10 °C min$^{-1}$ without equilibration in ambient; then repeat scans at 5, 2, 1 and 10 °C min$^{-1}$ sequentially after 1-h equilibration in the ambient (22 °C, 65–70% RH) in-between scans. **b** Derivative thermograms of (**a**). **c**, **d** Segmented TG of mTFF-SO$_3$-Na powder, and mTFF-C$_2$F$_5$SIS-Na powder, respectively. Legend for (c) and (d): solid line, first scan; dashed line, second scan, each after equilibration in the ambient. All thermograms were conducted in flowing nitrogen. Water desorption before first data point causes it to start below 100% mass. The hydration number $n_w$, defined as H$_2$O per ion pair, is given for each step.

PVB-NMe3-Cl). The 'universal' desorption characteristics observed suggest a simple labelling scheme. The water fraction that desorbs below 50 °C into dry nitrogen is labelled type-I, and those requiring heating between 50–150 °C and between 150–230 °C, type-II and type-III water, respectively (Fig. 2a).

## Type-I water

This type of water desorbs readily into dry nitrogen, and resorbs readily from humid ambient. Thus, it is partially lost during the TG setup. To quantify its amount accurately, we performed quartz-crystal-microbalance (QCM) measurements of films cast on the microbalance mounted in a humidity chamber. In this way, we found for PSSNa that $n_w$ for type-I water is 2.2, 4.9 and 7.3 H$_2$O/ion pair, at a water activity $a_w$ (equivalently, RH) of 0.58, 0.85 and 0.975, respectively, all at 22 °C (Supplementary Table 1: S/N 1). This agrees well with literature[58]. Likewise, mTFF-SO$_3$-Na gives type-I $n_w$ of 2.7, 5.2 and 7.3 for the same set of $a_w$ values. In fact, $n_w$ shows the same scaling behaviour with $a_w$ for all polyelectrolytes studied here. It roughly doubles as $a_w$ increases from 0.58 to 0.85, and triples as $a_w$ increases to 0.975 (Fig. 2b). Thus, the sorption isotherms are concave upwards[34,59–62]. The Flory–Huggins fit gives an apparent polymer–water interaction parameter of 1.0 (±0.1).

This simple result suggests that a simple scale can be set up to quantify the hygroscopicity for type-I water. If we choose PSSNa to define an index value of '1' and a hypothetical non-

hygroscopic material 'zero' on this scale, conjugated polyelectrolytes span a wide range, from 0.25 to more than 1. Hygroscopicity decreases with decreasing M$^+$ charge density[28,29,34,59,63], e.g. for TFB-CF$_3$SIS-M: Li$^+$ > Na$^+$ > Cs$^+$ > TMA$^+$. Similarly, it decreases with decreasing X$^-$ charge density, e.g. for mTFF-X-Na: $-$CH$_2$SO$_3^-$ > $-$CH$_2$SO$_2$N$^-$SO$_2$CF$_3$ ≈ $-$CH$_2$SO$_2$N$^-$SO$_2$CF$_2$CF$_3$.

## Type-II water

This type of water desorbs into nominally 'dry' nitrogen only at elevated temperatures. It can desorb at room temperature into vacuum, provided the vacuum is better than about 10$^{-5}$ mbar. Thus, type-II water can resorb from dry nitrogen at room temperature, and persist at $a_w$ of nominal zero. As $a_w$ decreases to this zero, the total $n_w$ falls to the type-II value, which sets the background on which the type-I water sorption isotherm builds. Indeed neutron reflectometry found sorbed water in dry polyelectrolyte multilayer films[38,64], and X-ray photoelectron spectroscopy found water even under ultrahigh vacuum conditions[53]. This phenomenon can be described by the Langmuir–Henry dual-mode model[65] and the Guggenheim–Anderson–De Boer model[66]. The amount of type-II water sorbed appears to vary with the type-I hygroscopicity index (Fig. 2b). It also decreases with decreasing cation and anion charge density: Li$^+$ > Na$^+$ > Cs$^+$ ≈ TMA$^+$ ≈ TEA$^+$ ≈ TPP$^+$ (Supplementary Table 1: S/N 9–14; 16 *cf* 17), and: $-$CH$_2$SO$_3^-$ > $-$CH$_2$SO$_2$N$^-$SO$_2$CF$_3$ ≈ $-$CH$_2$SO$_2$N$^-$SO$_2$CF$_2$CF$_3$ (S/N 6 *cf* 10, 15; 3 *cf* 16, 18).

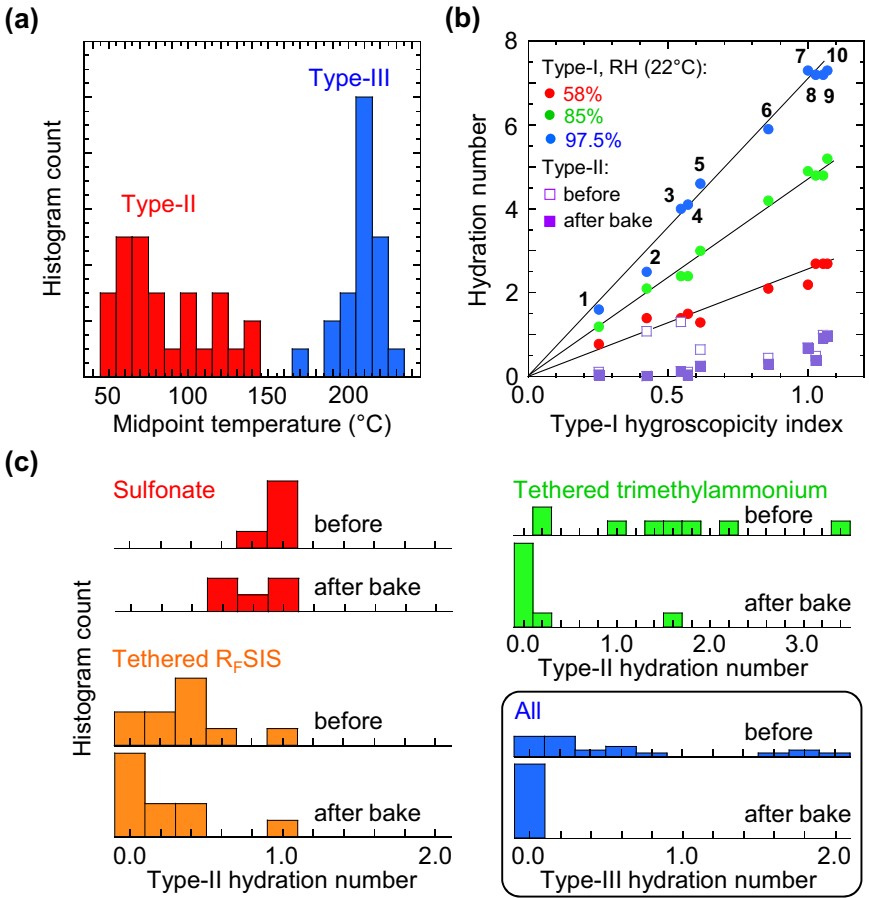

**Fig. 2 | Characteristics of hydrated polyelectrolytes. a** Histogram of desorption midpoint temperatures for type-II and type-III water. **b** Plots of type-I and type-II hydration numbers ($n_w$, in $H_2O$/ion pair) against type-I hygroscopicity index, defined to be 1 for PSSNa. Type-I $n_w$ are shown for three relative humidity (RH) values; type-II $n_w$ is independent of RH. Polyelectrolytes: 1, TFB-CF$_3$SIS-TMA; 2, TFB- NMe$_3$-TfO; 3, *p*-doped mTFF-C$_2$F$_5$SIS-Na; 4, TFB-CF$_3$SIS-Cs; 5, TFB-CF$_3$SIS-Na; 6, mTFF-CF$_3$SIS-Na; 7, PSSNa; 8, mTFF-C$_2$F$_5$SIS-Na; 9, TFB-CF$_3$SIS-Li; 10, mTFF-SO$_3$-Na; chemical structures in Supplementary Table 1. **c** Histograms of type-II and type-III $n_w$ for different polyelectrolyte families, before and after bake at 230 °C. All data-sets from Supplementary Table 1.

A surprising result is that polyelectrolytes with large hydrophobic cation and anion can still host a sizeable amount of type-II water. Examples include tethered trimethylammonium polyelectrolytes with X$^-$ ∈ {TfO$^-$, TFSI$^-$ and BArF$^-$} (S/N 22–26, 29, 30) and tethered R$_F$SIS polyelectrolytes with M$^+$ ∈ {TMA$^+$, TEA$^+$ and TPP$^+$} (S/N 12–14, 17). Treating these materials at high temperature, however, suppresses the resorption of type-II water, in some cases, for the largest ions, completely (Fig. 2c). Thus, a fraction of the binding sites for type-II water can be irreversibly eliminated by baking, which is required to turn these polyelectrolytes non-hygroscopic.

## Type-III water

This type of water typically shows only irreversible desorption (Fig. 2c). Its amount becomes large when water is present during processing (S/N 3, 5, 7, 8). For example, when TFB-SO$_3$-Na is precipitated from dimethyl sulfoxide solution using water instead of chloroform, the $n_w$ for type-III water jumps from 0.12 to 0.66 $H_2O$/ion pair (S/N 5 *cf* 6). Thus, in most of the work here, we use diethyl ether as the precipitating non-solvent. Even so, tethered trimethylammonium polyelectrolytes can still host large amounts of type-III water (S/N 22–26, 29, 30), whose binding sites can be fully eliminated by heating.

Overall, the results show that polyelectrolytes generally exhibit a diversity of behavior in hydration and a larger hygroscopicity than even hydrophilic ionic liquids. In the moderately hydrated regime, they host at least three distinct sorbed water species. Clear, the sorption sites are associated with the ion clusters, since the polymer backbones themselves do not sorb much water. For example, both poly(3-hexylthiophene) and tetradecyl-substituted poly(bithiophene-*alt*-thienothiophene) sorb only *ca.* 0.005 $H_2O$/thiophene ring in the ambient[17]. Some of these sites, however, can be eliminated by heat treatment.

## Desorption activation energy analysis

To identify the water binding sites, we first estimate the water binding energy by desorption kinetics analysis[67]. Figure 3 presents the results for the mTFF-SO$_3$-Na sample. We evaluated the surviving water fraction $\alpha$ as a function of time $t$ in dry nitrogen at temperature $T$ of 25, 100 and 200 °C to study type-I, -II and -III water, respectively (Fig. 3). To evaluate the apparent kinetic order $n$, we fitted the data to Arrhenius kinetics: $-\frac{d\alpha}{dt} = A \exp(-\frac{E_a}{k_B T})\alpha^n$, where $A$ is the pre-exponential factor, $E_a$ is the activation energy, and $k_B$ is the Boltzmann constant. The ln (–d$\alpha$/d$t$) *vs* ln ($\alpha$) plots turn out to be linear, giving $n \approx 1.8$ for type-I water (Fig. 3a), and 1.0 for both type-II and -III water (Fig. 3b, c). The pseudo second-order kinetics indicates that the desorption rate of type-I water decreases as $\alpha$ decreases, suggesting interaction between the water molecules. The unimolecular kinetics, on the other hand, indicates that the desorption rates of type-II and -III water are constant, suggesting the water molecules bind to quasi-independent sites.

To evaluate $A$ and $E_a$ for type-II and -III water, we thus fitted their segmented TG data to first-order kinetics: $\ln\left(-\frac{d(\ln\alpha)}{dt}\right) = \ln A - \frac{E_a}{k_B T}$. This

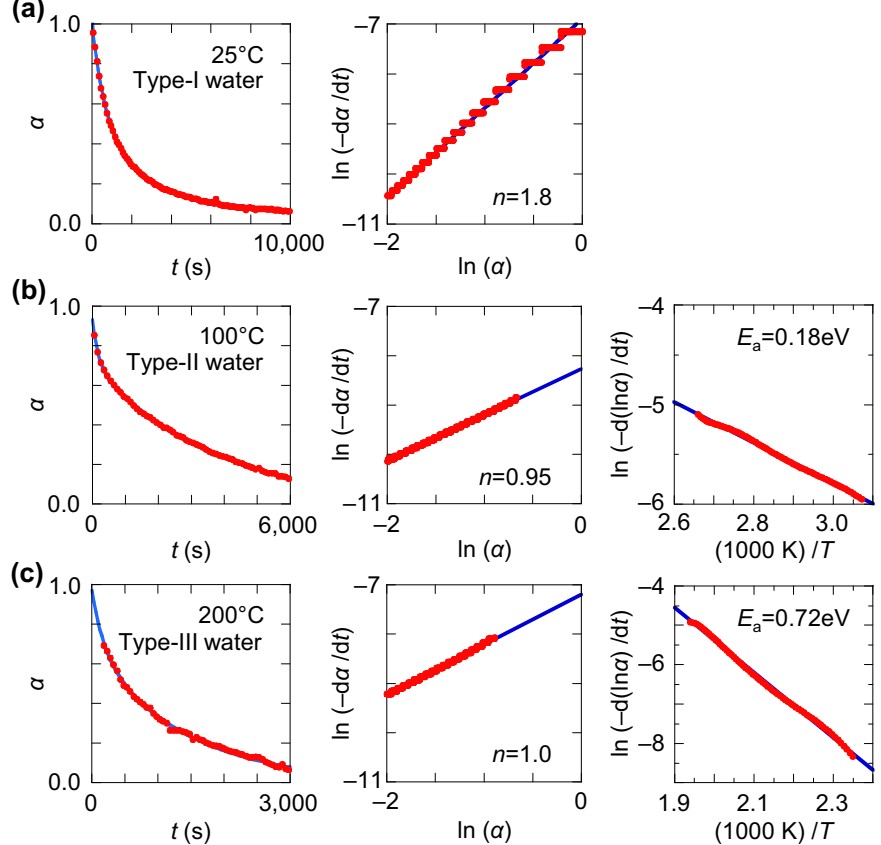

**Fig. 3 | Desorption kinetics analysis for an mTFF-SO3-Na sample.** Plots of remaining water fraction $\alpha$ against time $t$, plots of $\ln(-d\alpha/dt)$ $vs$ $\ln(\alpha)$; and plots of $\ln(-d(\ln\alpha)/dt)$ $vs$ $1/T$, for **a** type-I, **b** type-II, and **c** type-III water, to extract the apparent kinetic order $n$ and activation energy $E_a$. Symbols, data; lines, guide-to-the-eye or linear fits.

gives $E_a$ of 0.18 and 0.72 eV, respectively. The $E_a$ and $A$ parameters are compiled in Supplementary Table 2. These parameters characterize the desorption–temperature profile. A steeper step gives a larger $E_a$ together with a larger $A$. Such a correlation is reminiscent of the enthalpy–entropy compensation[68]. But it may also receive a contribution here from irreversibility. Assuming diffusion through the sample is not rate-limiting, $E_a$ should reflect the water binding energy[69,70].

The results reveal that $E_a$ values for reversible desorptions are generally lower than for irreversible ones. For reversible type-II water, the $E_a$ values fall in the range of 0.15–0.2 eV for tethered sulfonate polyelectrolytes (Supplementary Table 2: S/N 1–5) and 0.2–0.3 eV for tethered $R_FSIS$ polyelectrolytes (S/N 6–9), largely independent of the cation. They are $ca$. 0.2 eV for tethered trimethylammonium polyelectrolytes (S/N 15, 19). For irreversible water, the $E_a$ values can be considerably larger. For irreversible type-II water, the $E_a$ values are 0.3–0.4 eV for the usual polyelectrolytes (S/N 11–14, 16, 17) and 0.55–0.65 eV for those with large hydrophobic ion pairs (S/N 10 and 18). For type-III water, the $E_a$ values lie between 0.35 eV and 0.95 eV. Given the hydrogen-bond (H-bond) binding energy in a water dimer is only 0.15 eV[71,72], the results suggest that reversible type-II water is bound with similar or marginally larger strengths, but irreversible water is bound much more strongly, with a strength approaching those of multivalent and transition metal cations[73].

## Water binding motifs: OPLS4 simulations

To survey the possible binding motifs, we employ the OPLS4 all-atom molecular mechanics force field. This has been well-parametrized for a wide chemical space[74,75]. It has also been found to have acceptable accuracy for electrostatic and H-bond simulations (Supplementary Table 3). We computed various hydrated ion clusters given by the formula $(2\,M^{+-}A–(CH_2)_7–A^-)_8\,(H_2O)_n$, where $A^- \in \{SO_3^-,\ C_2F_5SIS^-\}$, $M^+ \in \{Na^+,\ TMA^+\}$, and $n \in \{24,\ 104\}$, corresponding to $n_w$ of 1.5 and 6.5, respectively, with the $–(CH_2)_7–$ tether imposing realistic steric constraints to simulate the ion cluster 'surface'.

Molecular snapshots of the thermally-equilibrated clusters reveal three essential features. First, the cation–anion framework is not disrupted by hydration at the level of one to two water molecules per ion pair. The water molecules do not insert into the ion clusters, but bind to ions at their surfaces. Even at a moderately high $n_w$ of 6.5, the cation–anion framework is still substantially retained, but water intrusion splits the ion cluster into smaller aggregates. See Fig. 4, and Supplementary Video 1, which provides 360°-view of the hydrated clusters. [This tendency, however, may have been amplified by overestimation of water-ion binding energies in OPLS4.] Second, the distribution of water is inhomogeneous. At $n_w$ of 1.5, most of the water molecules bind to the ion cluster directly in what we term as "primary hydration". At $n_w$ of 6.5, many of the water molecules bind to other water molecules, in what we term as "secondary hydration".

Third, the primary hydration water can bind in several motifs via: (i) H-bond to the anion, which we denote $\alpha$ binding ('$\alpha$' for anion); (ii) H-bond between two adjacent anions, $\beta$ binding ('$\beta$' for bridging); (iii) electrostatic bond to the cation, $\chi$ binding; (iv) dual bond to both cation and anion, $\delta$ binding; or (v) combination bond to one cation and two anions, i.e., $\delta + \alpha$ binding. This contrasts with the ionic liquid picture, where the water molecules participate primarily in $\beta$ binding which then elaborates into H-bonded chains and clusters in the bulk of the ionic liquid[47–50].

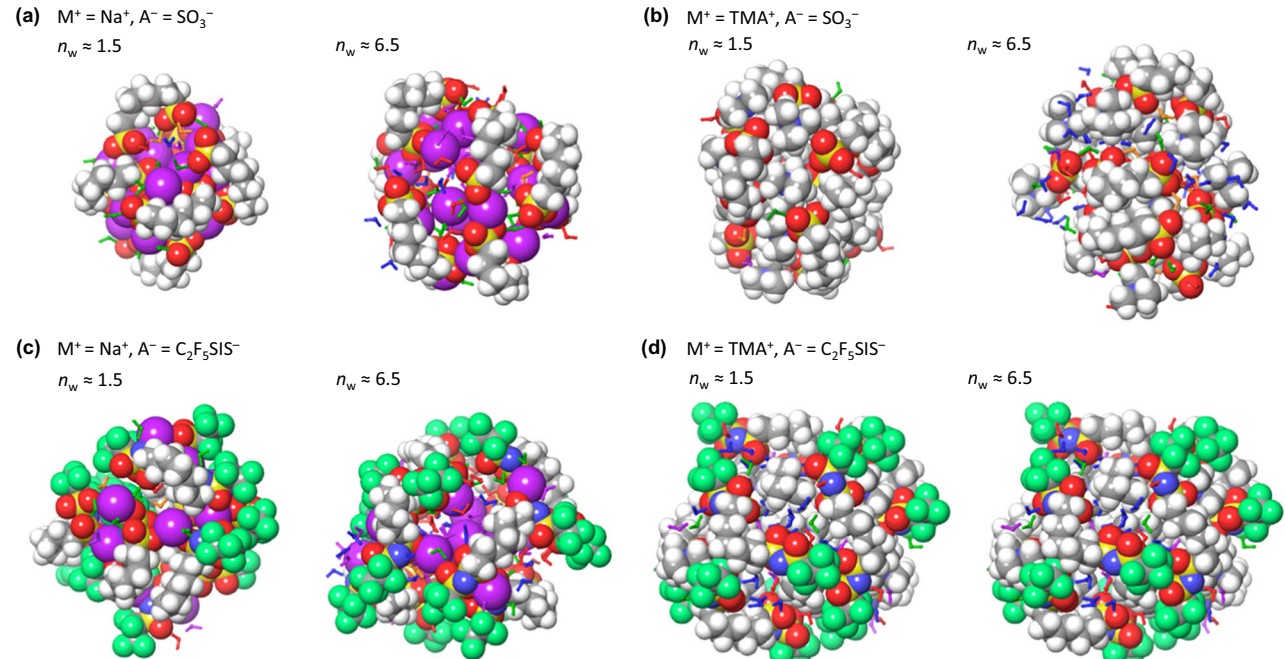

**Fig. 4 | OPLS4 molecular snapshots of $(2M^+\cdot A^-\cdot(CH_2)_7\cdot A^-)_8$ $(H_2O)_n$ hydrated ion cluster models for $(M^+, A^-)$. a** $(Na^+, SO_3^-)$, **b** $(TMA^+, SO_3^-)$, **c** $(Na^+, C_2F_5SIS^-)$, and **d** $(TMA^+, C_2F_5SIS^-)$. Atom legend: red, O; yellow, S; white, H; grey, C; violet, Na; blue, N. Atoms are shown in space-filling view to visualize the cation–anion framework at the ion cluster surface; water molecules in skeleton view and coloured for different

binding motifs. Legend: red, α-binding, i.e., bonded to anion; orange, β-binding, i.e., bridging two anions; green, δ-binding, i.e., bonded to cation–anion pair; violet, χ-binding, i.e., bonded to cation; blue, water molecule in secondary hydration. MD production run: 4 ns at 300 K.

Thus, OPLS4 indicates a preference for surface rather than internal hydration of the ion clusters in polyelectrolytes. This is attributed to the excess free energy and volume present at such surfaces. Because the sorbed water cannot attain the 3D order necessary for freezing, it becomes non-freezable. Thus, non-freezability alone does not imply strong binding.

**Water binding energies: DFT computations**

We employ DFT/CAM-B3LYP/6-31++G(*d,p*) to compute the water–site binding energies and associated vibrational frequencies. This level of DFT theory reproduces the H-bond better than OPLS4. See Supplementary Table 3, and also Supplementary Tables 1-1, 1-2 and 2 in ref. [52]. Due to computational costs, however, the calculations are limited to small ion multiplet models. We studied $(M^+ X^-)_r (H_2O)_p$, where $M^+ \in$ {$Li^+, Na^+, K^+, TMA^+$}, $X^- \in$ {$RSO_3^-, TfO^-, (CH_3SO_2)(CF_3SO_2)N^-$}, and $r \in$ {1, 2, 4}.

Figure 5 shows selected geometry-optimized structures, while Supplementary Figs. 2, 3 and 4 show the structures for $Na^+$, $Li^+$ and $K^+$, and $TMA^+$, respectively, totalling sixty. The computed water binding energies $\Delta U_o$ are compiled in Supplementary Table 4. Such values should be regarded as upper limits because the optimal geometry for the gas phase may be frustrated in the condensed phase. Moreover, because small ion multiplets expose cations more strongly, the calculations yield more occurrences of χ, δ and δ + α binding.

The results reveal that δ binding is strong. The $\Delta U_o$ values are a few tenths of an eV, varying strongly with cation. Representative δ motifs are shown in Fig. 5 (for ion pairs: **II** b; **IV** b & d; and ion octets: **II** i & j; **VI** i & j; and **VII** f). The binding energies decrease by 30% as $r$ increases from 1 to 4 for sulfonate clusters with $M^+ \in$ {$Li^+, Na^+, K^+, TMA^+$}. This decrease stems from an averaging out of the electrostatic field gradient at the binding site, which is a manifestation of the local Madelung potential effect[76,77]. Since realistic ion cluster sizes are generally larger than four[26,27],

realistic δ binding energies are estimated to lie in the range of 0.35–0.55 eV.

The α binding energies are smaller. But they vary little with the anions studied. $\Delta U_o$ is *ca.* 0.3 eV for sulfonate, and 0.25 eV for both TfO⁻ and $(CH_3SO_2)(CF_3SO_2)N^-$. Typical α motifs are given in Fig. 5 (**II** a) and Supplementary Figs. 2–4 (**I** a, **I** e–g, **IV** a, **VI** a and **IX** a). β Binding gives similar $\Delta U_o$ values as α binding, not twice as large as might have been naively expected. Typical motifs are shown in **V** g, **V** i and **V** j. Finally, χ binding varies strongly with cation: 0.45 eV for $Li^+$, 0.65 eV for $Na^+$, and 0.15 eV for $TMA^+$. Typical motifs are shown in **I** b, **III** c, **V** h and **VII** g.

The electrostatic interaction with cation can be modelled as a water–cation dipole interaction. Such an effect persists beyond physical contact of the water molecule with the cation. To show this, we employ PM3 to compute $\Delta U_o$ for δ binding in a variety of cation–anion pairs (Fig. 6), and find: $\Delta U_o = A + B * d^{-2}$, where $d$ is the water oxygen-to-cation distance, $A$ is the limiting binding energy (0.25 eV for $CH_3SO_3^-$, and 0.20 eV for both TfO⁻ and TFSI⁻), and $B$ is an empirical constant (2.5 eV Å²). The form of this equation is characteristic of the dipole–ion interaction. It predicts the attraction to remain sizeable ($\gtrsim$0.1 eV) for distances up to 5 Å. This implies a proximal cation not in direct contact with the water molecule can still contribute to electrostatic bonding that reinforces its α binding.

In summary, the computed results here provide the basis to assign binding motifs to the measured $E_a$ values. Values in the range of 0.15–0.3 eV are consistent with α and β bindings, possibly reinforced with electrostatic bonding to a proximal small cation or to a contacting large cation. Values in the range of 0.3–0.6 eV are consistent with χ, δ and δ + α bindings; but values larger than 0.6 eV must involve caging effects. Thus, reversible type-II water is chiefly H-bonded to anions, but irreversible type-II water involves strong contribution also from the cation, while type-III water involves additional caging effects.

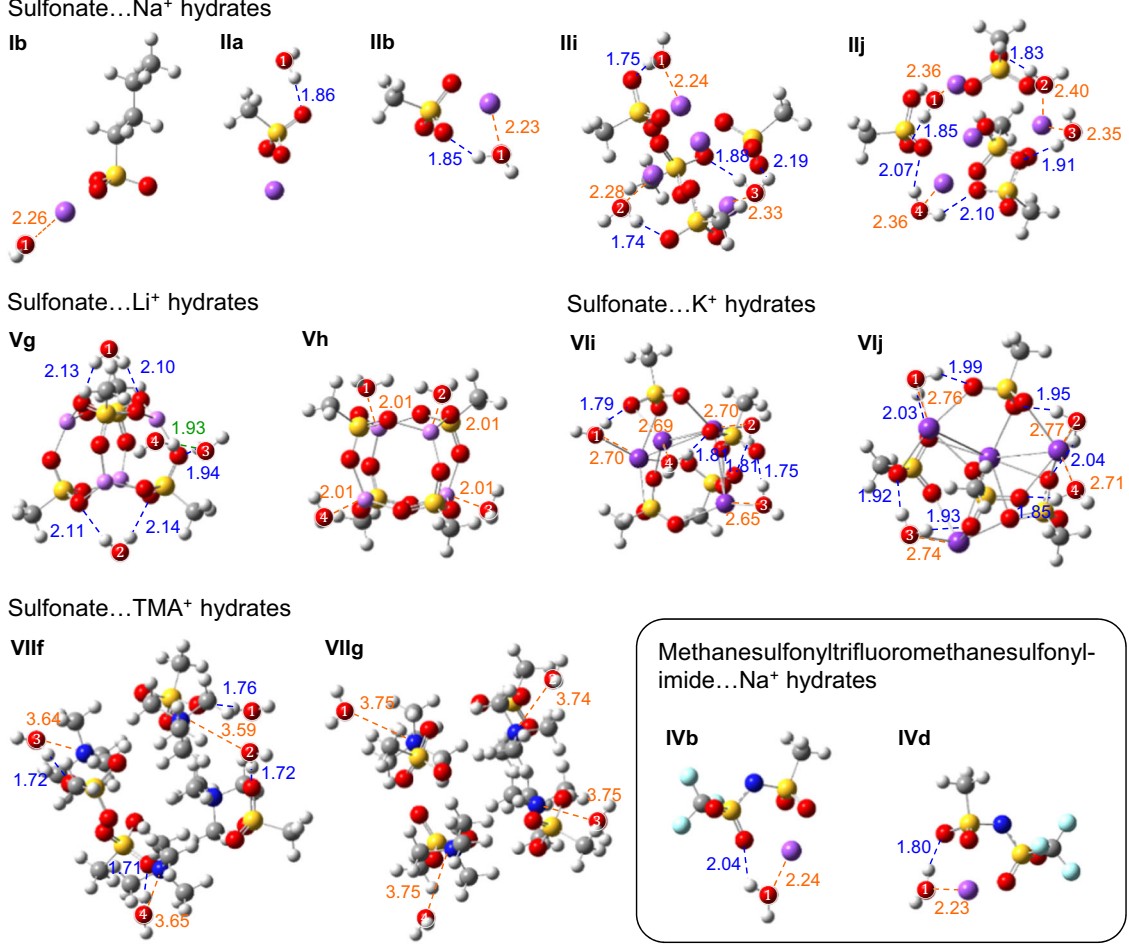

**Fig. 5 | Selected water binding motifs in $(M^+ X^-)_r (H_2O)_p$ hydrated ion multiplets.** Methodology: Geometry optimization and energy minimization by DFT/CAM-B3LYP/6-31++G($d,p$). Bonded distance given in Å from water to: anion (blue dashed line), cation (orange) and H-bonded water (green), with water molecules numbered. Atom legend: violet, Li, Na or K; red, O; yellow, S; blue, N; cyan, F; grey, C; white, H. Full set in Supplementary Figs. 2–4. The H...O distance in H-bonded water dimer is 1.89 Å at this level of theory.

## Hydrogen-bond character: computed frequencies

To confirm these assignments, we compute the $\nu$OH wavenumbers and compare them with the observed wavenumbers (Supplementary Table 4)[78]. The $\nu$OH vibration is highly sensitive to binding motif of the water molecule, in particular, the amount of H-bond donor character (D) and H-bond acceptor character (A) at the O–H group of the motif[79–81]. To extract association insights, we first generate a correlation chart between $\nu$OH wavenumber and H-bond character, based on literature calculations of water clusters (Fig. 7)[79]. The chart shows that $\nu$OH of water shifts systematically with A and D character, as is well known[79,80,82–84].

Then we added to this chart our calculated results for the hydrated ion multiplets. The augmented chart shows that $\nu$OH of a water molecule changes little between H-bonding (as donor) to another water or to sulfonate (black $cf$ red bands). This has been confirmed by measurements in ionic liquids[44,46]. An additional electrostatic bonding to $M^+$, however, behaves like adding weak A character to the water molecule, redshifting its $\nu$OH band. The amount of redshift varies little with the size of $M^+$. Thus, $\alpha$ binding under the influence of a proximal small cation produces a similar spectroscopic signature as $\delta$ binding, corresponding to DA with weak A character. Similarly, $\beta$ binding under the influence of a proximal small cation produces a similar spectroscopic signature as $\delta + \alpha$ binding, corresponding to DDA with weak A character. In this way, the observed $\nu$OH

wavenumber can be interpreted to support or exclude certain binding motifs.

## Hydrogen-bond character: experiments

To realize this analysis for mTFF-SO₃-Na, we collected in situ FTIR spectra of its film in the following sequence: ambient air (where $n_w \approx 5$) → nitrogen ($n_w \approx 1$) → after 230°C-bake, in nitrogen ($n_w \approx 1$) → after short exposure in ambient air ($n_w \approx 2.5$). The spectra in ambient air contain both type-I and -II water, while that in nitrogen only type-II water. These spectra and their difference (Fig. 8a, b, respectively) show that hydration intensifies the $\nu$OH envelope at 3460 cm⁻¹, and redshifts the SO₃ bands, both $\nu_s$ (symmetric) and $\nu_a$ (anti-symmetric), as expected[85,86]. The $\nu$OH envelope is considerably blue-shifted from bulk water[87].

We modelled this envelope with a sum-of-Gaussians: $I = \sum_i I_i \exp\left(-\frac{(\nu - \nu_i)^2}{2\sigma_i^2}\right)$, where $\nu_i$ is the wavenumber and $\sigma_i$ is the standard deviation, of the $i$-th component (Fig. 8c). The component basis set employed corresponds approximately to the following H-bond character (in cm⁻¹): (i) 3200 – DAA; (ii) 3300 – DAA, and DA with strong A; (iii) 3420 – DAA, DDAA, and DA with weak A; (iv) 3520 – DDA, and D; and (v) 3630 – DD, and free OH; where "weak A" captures electrostatic interaction with $M^+$, and "free OH" refers to that in the condensed phase, not vacuum. The 3630-cm⁻¹ component is significantly

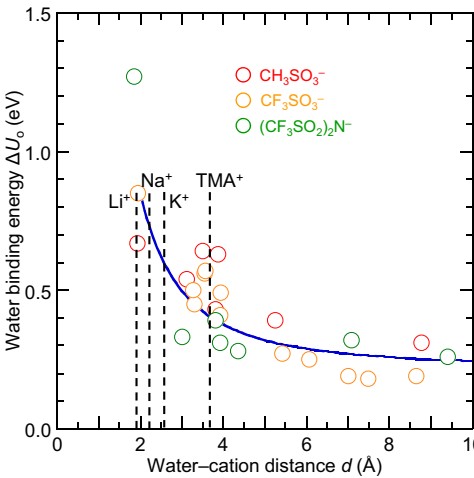

**Fig. 6 | Water binding energies to cation–anion pairs at PM3 level.** Water molecule singly H-bonded to anion {$CH_3SO_3^-$ (red), $CF_3SO_3^-$ (yellow), $(CF_3SO_2)_2N^-$ (green)} under influence of cations {$Li^+$, $Na^+$, $K^+$, $TMA^+$, $TEA^+$ and $TPP^+$} giving different water oxygen–cation distances $d$. Methodology: Difference of internal energies of formation before and after removal of the water molecule, for geometry optimization at PM3 level. Blue line is empirical fit to: $\Delta U_o = A + B * d^{-2}$, with $A = 0.22$ eV and $B = 2.5$ eV Å$^2$. The locations of water–cation contact distances given by DFT are marked by dashed lines.

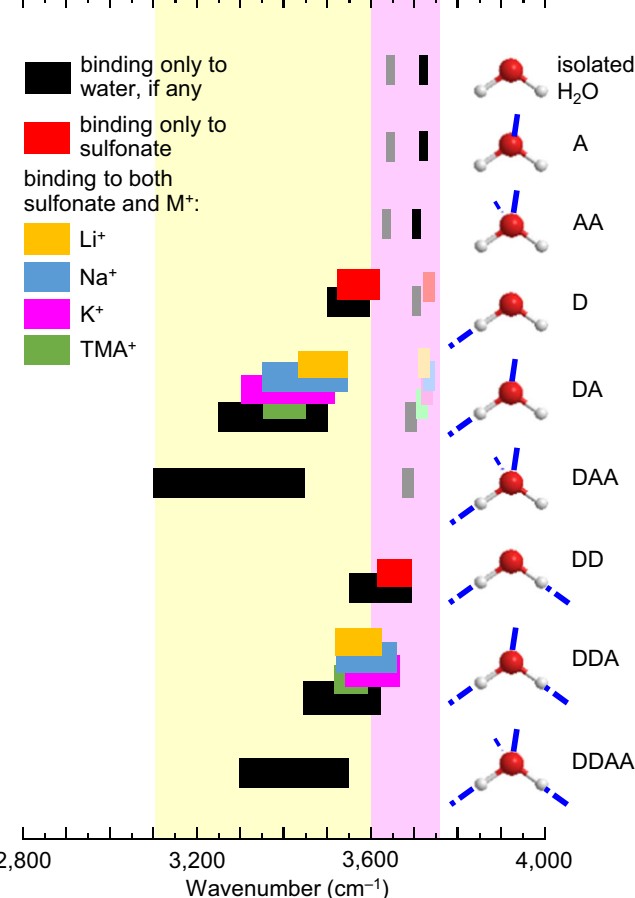

**Fig. 7 | Computed $v$OH wavenumber correlation chart for water molecules.** Hydrogen-bond character labels: 'D' denotes H-bond donor; 'A', H-bond acceptor. Wavenumber ranges: saturated colour and black bands for strong infrared absorptions; tinted colour and grey bands for weak absorptions. Water cluster data from Ref. [79]. Water bonded to ion-multiplet data from present work: DFT/CAM-B3LYP/6-31 + + G($d$,$p$), with wavenumber scaling: $v = v' * (1.184 − 0.00006\,v')$, where $v'$ is the DFT frequency. The wavenumber of 'free' $v$OH redshifts significantly with van der Waals interaction in condensed phase. Yellow and magenta colourations are guide-to-the-eye.

narrower than the others ($\sigma_i$, 38 vs 60 cm$^{-1}$), so it corresponds indeed to free OH.

Type-I water yields very weak intensity in the 3200-cm$^{-1}$ component (spectra i and iii). This indicates that DAA character is nearly absent, which then also rules out other extensively H-bonded water species. Thus, type-I water comprises small H-bonded water chains and clusters with DA, DD, D and/or A character, together with free OH. Type-II water yields strong intensities in the 3420; 3520 and 3630-cm$^{-1}$ components (spectrum ii). This indicates that DA (weak A) character, and possibly D, is present together with free OH, which is consistent with α binding under the influence of proximal cation and/or δ binding. We can rule out any significant β binding, or δ + α binding, as these show two sharp bands over the 3525–3675 cm$^{-1}$ spectral region[44,46]. Since $E_a$ analysis has ruled out δ binding for mTFF-SO$_3$-Na, the dominant motif for its type-II water is α binding under the influence of proximal cation.

To estimate population size, we collected in situ Raman spectra. Raman cross sections vary much less strongly with H-bond character than infrared cross sections[88,89]. Thus, Raman spectroscopy is more suitable for quantification. The observed Raman spectral components are broadly consistent with the infrared components (Fig. 8d). Taking into account the approximate Raman cross sections—equal for all H-bonded OH and halved for free OH[88,89]—we deduce:

a. ratio of type-I to type-II water is roughly five, which agrees with QCM measurements;
b. type-I water has a bonded OH-to-free OH ratio of roughly 2:1, which confirms small H-bonded water chains or clusters[90–92];
c. type-II water has a bonded OH-to-free OH ratio of roughly 1:1, which agrees with the assigned α binding.

**Summary analysis of a hydrophobic polyelectrolyte family**
We applied a similar analysis to the TFB-CF$_3$SIS-M family of polyelectrolytes. Their hygroscopicities are lower than mTFF-SO$_3$-Na, and decrease with increasing size of $M^+ \in$ {$Li^+$, $Na^+$, $Cs^+$, $TMA^+$, $TEA^+$} (Fig. 9a). Type-II water (red dotted line) is dominated by three components at 3350, 3500 and 3630 cm$^{-1}$, largely independent of $M^+$ (Fig. 9b). This indicates predominant α binding with D and/or DA (weak A) character, together with free OH. H-bonding of water to CF$_3$SIS results in a smaller redshift of its $v$OH by 50–100 cm$^{-1}$ compared with sulfonate (Supplementary Table 4; see also refs. [44,46]). On the other hand, type-I water is dominated by components at 3520 and 3630 cm$^{-1}$, which indicates D character and higher free OH content than mTFF-SO$_3$-Na. Thus, the type-I water in TFB-CF$_3$SIS-M occurs in even smaller H-bonded clusters. Its desorption shifts $v_s$ SO$_2$ bands (1060 and 1200 cm$^{-1}$) towards lower wavenumbers for $M^+ \in$ {$Li^+$, $Na^+$} but higher wavenumbers for {$TMA^+$, $TEA^+$}. Again, type-I water influences anion vibrations, mediated perhaps by type-II water, similar to the case of sulfonate polyelectrolytes.

In summary, the reversible type-II water in both hydrophilic and hydrophobic polyelectrolyte models here comprises water molecules that are chiefly H-bonded to the anion with weak electrostatic bonding to the cation. On the other hand, type-I water comprises small H-bonded chains and clusters attached to type-II water. Thus, the pronounced effect of cations on hygroscopicity is attributed to proximal effect, and influence via anion, packing and cluster morphology, rather than direct bonding to water.

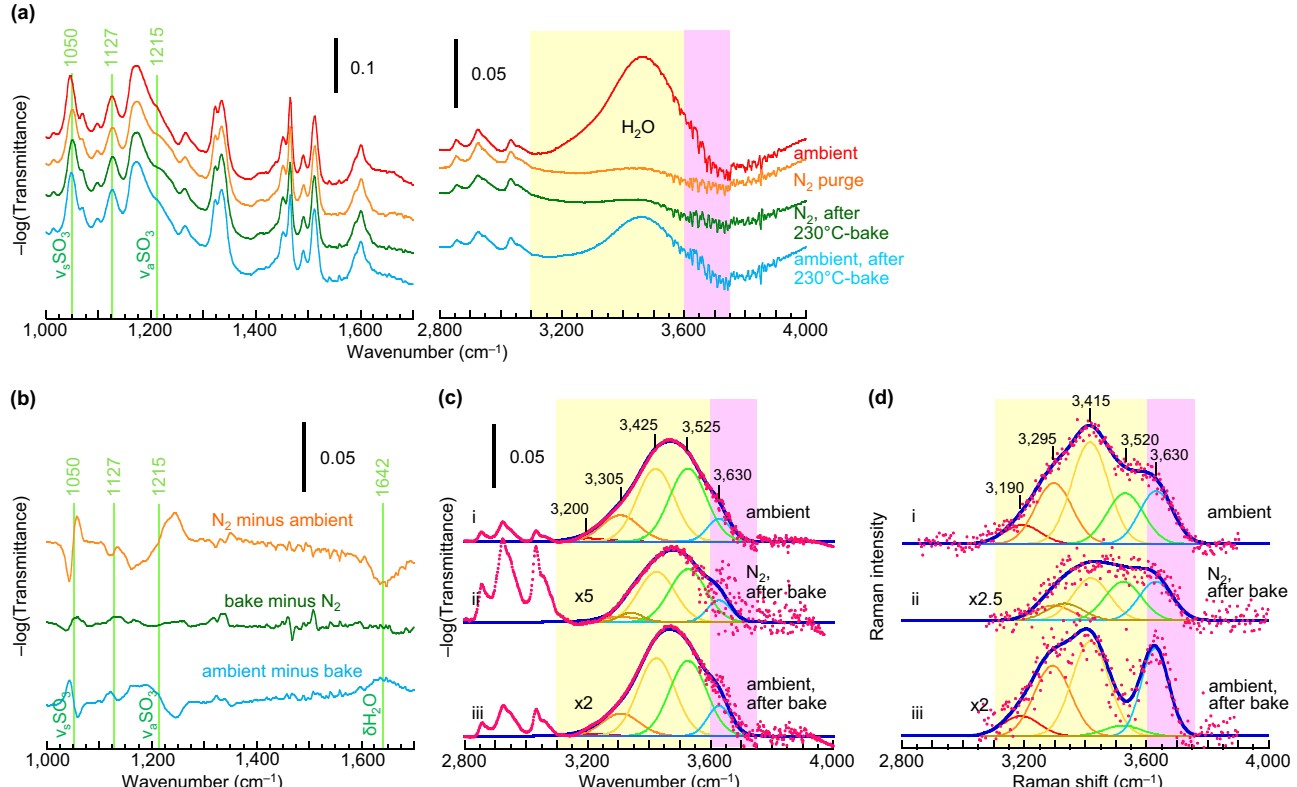

**Fig. 8 | Vibrational spectroscopy of mTFF-SO₃-Na film undergoing a dehydration−rehydration cycle. a** Transmission FTIR spectra collected in an optical-access flow-through cell: (red) after exposure to ambient air, (orange) in situ after 15 min in flowing nitrogen, (green) after 15-min bake at 230 °C in nitrogen and cooling to room temperature, and (blue) in situ after 5-min exposure to ambient air. **b** Difference spectra in the 1000−1700-cm⁻¹ region. **c** Gaussian curve-fitted FTIR *v*OH envelope of (**a**). **d** Gaussian curve-fitted Raman *v*OH envelope. Raman spectroscopy was performed in backscattering geometry on Si wafer mounted on hot stage, with excitation wavelength of 514 nm. Films were cast from acetonitrile solution. Atmospheric contribution by H₂O and CO₂ to FTIR spectra, and fluorescence contribution by glass to Raman spectra, were removed by subtraction.

## Generalization: internal vs surface hydration models

The internal hydration and surface hydration models are illustrated schematically in Fig. 10a, b, respectively, to underscore fundamental differences in their nature and binding of water. Evidences from desorption activation energy analysis and *v*OH vibration analysis support the surface hydration model for monovalent polyelectrolytes in the moderate hydration regime of up to several water molecules per ion pair. Reversible type-II water is substantially held by α binding to the anion at the ion cluster surface, reinforced by electrostatic bonding to proximal small cation or to contacting large cation, while keeping the ionic framework substantially intact. This is undoubtedly driven by electrostatic stabilization of the ion cluster core. Despite its greater strength, δ binding to anion–small cation pair appears to be generally suppressed, even at the ion cluster surface, despite OPLS4 predictions. This suggests recession of the small cation away from the surface and other local geometry effects that frustrate dual binding. The model explains the existence of "void water" and its orientational mobility. At higher hydration levels, water may intrude into the ion cluster to 'carve out' new surfaces. Type-I water is characterized by small H-bonded chains or clusters, not an extensively H-bonded network. On the other hand, irreversible type-II water is likely held by δ binding, possibly in a weak cage, also at the surface of the ion cluster. This becomes particularly prominent in polyelectrolytes with large hydrophobic cations. Type-III water is held by δ binding, often in a strong cage, likely in the interior of the ion cluster, due to trapping of molecular water during material deposition.

A key consequence of surface hydration of the ion clusters is that hygroscopicity, and hence its control, depends not only on hydrophilicity of the ion pairs, but also morphology of the ion clusters. The latter is determined by relative sizes of the ions, polymer backbone structure, ion attachment site, ion density, etc. There is evidence for this from the variation in hygroscopicity with different polymer backbone structures for the same ion pair (Supplementary Table 1). Another key consequence is that the hydrated complexes can be described as small H-bonded water clusters attached to the local ion packing. The water substantially retains its own properties−for example, ionization energetics, chemical reactivity−albeit perturbed by the interaction. This picture has enabled the influence of sorbed moisture on several phenomena to be successfully modelled, including: dehydration-induced electron transfer from small ion clusters[16], hydration-induced chemical trapping of holes from ultrahigh-workfunction polymer semiconductors[15,52], and hydration-induced interference of nitrene photocrosslinking mechanism[54,55].

Finally, we summarize here the hygroscopicity trends for charge-doped polymers. Charge-doping introduces mobile carriers on the polymer backbone, which are balanced by counterions, whether free or tethered. Similar to their undoped counterparts, charge-doped polyelectrolytes also show both type-I and -II water, with similar hygroscopicity (Supplementary Table 1: S/N 4 *cf* 3; 19 *cf* 18; 21 *cf* 20; 27 *cf* 26). But charge-doped polymers that are counterbalanced by particularly large hydrophobic ions can show considerably lower hygroscopicity (S/N 31 and 32) than those counterbalanced by presently available tethered ions. This suggests further reduction in the hygroscopicity of charge-doped

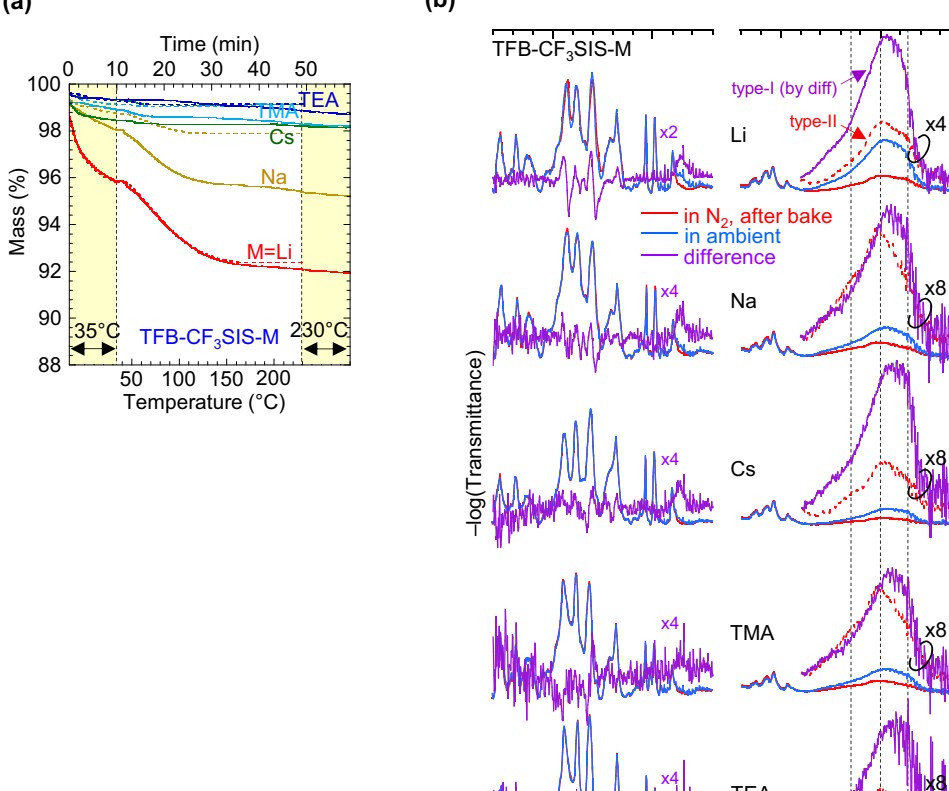

**Fig. 9 | Thermogravimetry and infrared spectroscopy of TFB-CF₃SIS-M.**
**a** Segmented TG of powders, at 5 °C min⁻¹. **b** FTIR spectroscopy of films baked in dry nitrogen, and after exposure to ambient air. Methodology: Films were baked on hotplate at 180 °C in nitrogen glovebox, and spectra collected in N₂ (red lines); then films were equilibrated in ambient air, and spectra collected in ambient air (blue). Difference spectra are shown in violet on expanded scale, together with the spectra in N₂ (red dashed lines). The vertical black dashed lines mark the *v*OH components for type-II water in the samples, at 3350, 3500 and 3630 cm⁻¹.

polyelectrolytes should be possible with better design of the tethered counterion.

## Methods

### Materials
Polyelectrolytes were synthesized and ion-exchanged following well-established protocols in our laboratory[15,26,27]. PSSNa and PVBTMA-Cl were commercial products used as received. Where required, films or solutions were doped following well-established protocols in our laboratory, as indicated in Supplementary Table 1[15,26,27]. All polyelectrolytes were dried at 80 °C overnight in a dynamic vacuum ($10^{-2}$ mbar), and stored in a nitrogen glovebox.

### Thermogravimetry and quartz crystal microbalance
Thermogravimetry was performed on Discovery TGA 1 (TA Instruments). Samples in powder form (4–6 mg) were loaded in the ambient (22 °C, 65–70% RH) onto the sample pan which was then moved into the nitrogen-purged furnace under computer control. Typical scan parameters: isothermal hold at 35 °C for 10 min, temperature ramp at 5 or 10 °C min⁻¹ to 230 °C, isothermal hold for 10 min, temperature ramp at −5 °C min⁻¹ to 35 °C, all in flowing nitrogen (>99.999% purity). The samples were typically exposed to the ambient for 1–2 h before repeating the scan. Quartz crystal microbalance measurements were performed on QCM200 Quartz Crystal Microbalance (Stanford Research Systems) at room temperature with a 5 MHz AT-cut quartz crystal. Films were spin-cast onto the

quartz crystal and mounted inside a gas flow cell. Resonant frequencies were measured in dry nitrogen ($f_o$) and in controlled-humidity nitrogen ($f$). The mass of water sorbed by the film ($\Delta m$) was computed by the Sauerbrey equation: $\Delta f = -C_f \times \Delta m$, where $C_f$ is the crystal sensitivity factor (56.6 Hz μg⁻¹ cm²) and $\Delta f$ is the change in frequency given by $f - f_o$. The mass of film was estimated from its thickness, measured by profilometry, and density, measured by flotation method.

### FTIR and Raman spectroscopies
FTIR spectra were collected on a nitrogen-purged Nicolet 8700 FTIR spectrometer operated in the temperature-stabilized clean-room to provide measurement stability of better than 1 mAU in the transmission mode. Films were cast onto intrinsic silicon substrates and mounted inside an optical access flow cell fitted with KBr windows. Unpolarised Raman spectra were collected using a Renishaw Raman microscope in the back-scattered geometry. Films were cast onto intrinsic silicon substrates and mounted on a Linkam hot stage inside an optical access flow cell fitted with a quartz window. Ar ion laser (514 nm) was focused through a long working distance objective lens (numerical aperture 0.55, 50×) for excitation.

### Molecular mechanics and quantum chemical calculations
OPLS4 calculations were performed in MacroModel (Schrödinger LLC). The hydrated ion clusters were generated haphazardly and

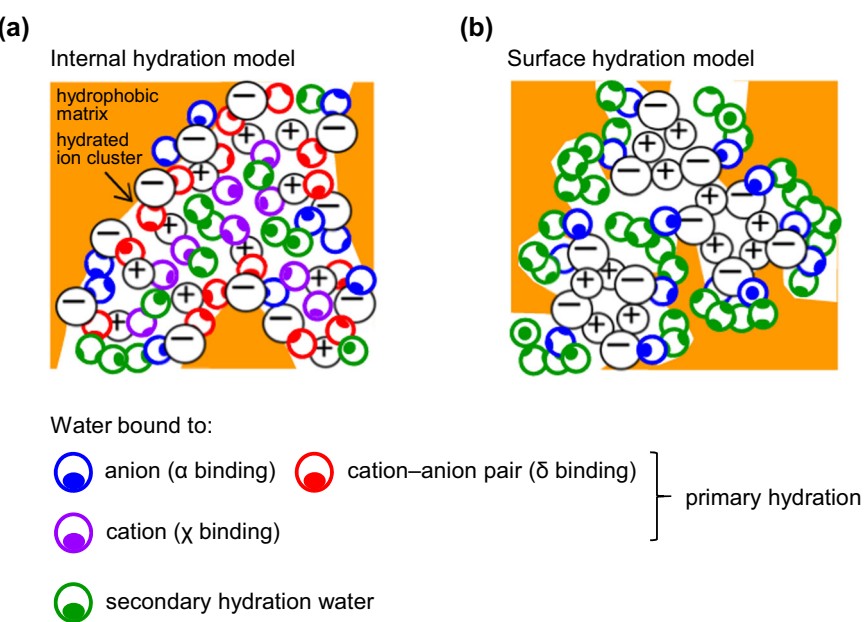

**(a)** Internal hydration model

**(b)** Surface hydration model

Water bound to:

○ anion (α binding)    ○ cation–anion pair (δ binding) ⎤
                                                        ⎥ primary hydration
○ cation (χ binding)                                    ⎦

○ secondary hydration water

**Fig. 10 | Schematic illustration of internal and surface hydration models for embedded ion clusters. a** Internal hydration model, and **b** surface hydration model. Cartoon is drawn for the moderate hydration regime. Circles with "–" represent anions, "+" represent cations; coloured circles with filled ellipses represent water molecules (where orientation of the ellipse suggests orientation of H atom). Water molecules are nearly spherical with van der Waals radius of 1.4 Å. For polyelectrolytes with large cation–anion pairs, α binding under the influence of proximal cation crosses over to δ binding.

equilibrated for 4 ns at 300 K, with background dielectric constant set to 1. Several snapshots were taken from the production run to check that the results are representative. DFT calculations were performed at the CAM-B3LYP/6-31++G($d,p$) level in Gaussian 09. The parent ion multiplet $(M^+ X^-)_r$ was first generated and geometry optimized. Then the desired number of water molecules were brought haphazardly to its surface to give a hydrated seed configuration, which was geometry re-optimized to give the local-minimum $(M^+ X^-)_r (H_2O)_p$ hydrated complex. Vibrational frequencies were computed. Then the water molecules were removed, and the geometry of the remaining $(M^+ X^-)_r$ re-optimized to give the corresponding dry complex. The adiabatic water detachment internal energy $\Delta U_0$ was evaluated as the difference in internal energies between the dry and hydrated complexes at 0 K, corrected for zero-point energy. Multiple hydrated seed configurations were generated to investigate the diversity of hydrated complexes.

## Data availability
Source data for all figures are available from the corresponding author.

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

## Acknowledgements

This research is partially supported by the National Research Foundation, Prime Minister's Office, Singapore under its Competitive Research Programme (CRP Award No. NRF-CRP24-2020-0006: A-0008375-00-00, A-0008375-01-00). The conjugated polyelectrolytes used in this report were designed and synthesized over few years by the combined effort of members of our chemistry team, including Qiu-Jing SEAH and Venu KEERTHI.

## Author contributions

C.G.T., M.N., M.M.S., M.C.Y.A., K.K.C. and Q.M.K. conducted the experiments. C.G.T., M.N. and R.Q.P. performed the quantum chemical calculations. R.Q.P. performed the OPLS4 calculations. M.C. and Y.P.F. performed the AMBER calculations. K.K.C. and M.C.Y.A. contributed to materials development. R.Q.P., L.L.C. and P.K.H.H. developed the research methodology and framework. L.L.C. directed materials development and characterization; R.Q.P. directed research project; P.K.H.H. directed theory support. L.L.C., R.Q.P. and P.K.H.H. wrote the manuscript. All authors discussed the experiments and results.

## Competing interests

The authors declare no competing interests.
