## [Peer Review File · Nature Communications]

REVIEWER COMMENTS

Reviewer #1 (Remarks to the Author):

The manuscript "Binding of molecular water in polyelectrolyte films" by Tang et al reports a detailed study on the hydration of polyelectrolyte films for a broad range of polyelectrolytes at low hydration levels. Using thermogravimetric desorption experiments, the authors find three different types of water molecules that are desorbed at different temperatures. Comparison of the desorption energy from experiment to quantum chemical calculations provides insights into the binding state of the water molecules. The inferred predominant binding to the anionic group of the polyelectrolyte is supported by density functional theory calculations of water's vibrational modes, which are compared to experimental infrared and Raman spectra. From these experiments and calculations the initial stages of polyelectrolyte hydration are shown to occur on the surface of ionic clusters, rather than the commonly assumed core of the ionic clusters. As such, the study provides new insights into polyelectrolyte hydration and the great number of studied samples shows that the derived hydration model is broadly applicable, which I find to be an extremely interesting and timely observation. Thus, in principle I can recommend publication in Nature Communications, however, based on the current manuscript some of the methodologies are not fully comprehensible to the reader and should be clarified before I can recommend publication. Details follow below:

- In figure 2 the authors show the TGA scans evidencing the three types of water. The authors argue that the three types have not been discriminated before. In the text it is implied that this is due to differences in the applied temperature profiles. In the present study the discrimination between type I and type II however relies on a plateau in the weight loss just before the temperature ramp is applied. As such, the loss after the first plateau may also be due to enhanced desorption due to increasing temperature. To provide unambiguous evidence the three types of water a more detailed discussion of the temperature profiles is necessary: How do the profiles look if no isothermal but a slowly increasing, continuous temperature ramp is applied? To what extent do the results depend on the scan rate?

- In Figure 4 the water binding energies as obtained from quantum chemical calculations are shown. Also here more details need to be given. As far as I understand, the hydrated ion-pair corresponds to a cation and an anion separated by one water molecule. This is however typically not the global energetic minimum for such calculations as the hydration energy is in the gas-phase much smaller than the attractive (electrostatic) contribution between the anion and the cation. According to my experience, direct cation-anion contacts are almost always energetically more stable in the gas phase. Thus, more details on how the geometries have been determined and optimized and how many conformations have been considered needs to be given. Also the phrase "ground state geometry" is not clear to me.

- Also for the vibrational spectra the analysis is somewhat unclear. The spectra in Figure 8b&c are presumably difference spectra, it is however not fully clear how these have been obtained. In Figure 8c the experimental data are shown only over a limited wavenumber range. To judge the reliability of the background subtraction the data should be shown over the same range as in 8a. Only such the reader can judge the significance of the fit with Gaussians. For these fits also the fitting equation should be given. Just stating a Gaussian width of 60cm^{-1} is ambiguous (e.g. square root of variance, or full width at half maximum).

- On page 9 (lines 232-236) the authors argue that the absence of a cation effect in E_a rules out direct cation contact. This should be toned down, as it just suggests that the potential energy due to interaction with the cation is not the dominant contribution. Further, the author argue that this rules out the core hydration model. This seems not supported by the experiments, as the energy does not tell anything about the location of the water. The author also have not calculated the hydration energy in the core, but just of an ion-pair, thus the data do not provide any insights into that.

- I find discussion of the IR results on page 12 difficult to follow. What are the main differences between the three different spectra in Figure 8c and d. What types of water supposedly prevails for the samples in different conditions? What is the inferred binding state for type I and type II water? Only referring to “(inspection, Figure 7)” makes it extremely difficult to follow the arguments.

- Also the sentence “The formation of extensive 3D H-bonded water clusters required in the core hydration model can also be definitively ruled out.” Is not clear. On what basis can it be ruled out? And why does the core hydration model require 3D H-bonded clusters at hydration levels as low as the once studied here?

Reviewer #2 (Remarks to the Author):

The authors of this paper use a combination of experiment (IR and Raman spectroscopy and thermogravimetry) and calculations (P3, DFT, D) to elucidate the nature of water interactions with polyelectrolytes. The authors provide strong evidence that the most tightly bound water molecules are surface bound rather than interior bound as has been assumed by most researchers in the field. This conclusion will strongly influence thinking in the field.

The paper is well written and the analysis carefully done. Sufficient information is provided to enable other researchers to reproduce their results. Publication is recommended.

Detailed Response to Referee Reports

Reviewer 1

We thank Reviewer for many suggestions and attention to details, which we have now adopted and addressed. As a result of the queries, we have added more computational and experimental data now, see revised and expanded Figures 2, 3, 5, 6, 8, 10, and new Table 3. Our line-by-line responses are as follows.

1. "... the discrimination between type I and type II however relies on a plateau in the weight loss just before the temperature ramp is applied. As such, the loss after the first plateau may also be due to enhanced desorption due to increasing temperature. To provide unambiguous evidence the three types of water a more detailed discussion of the temperature profiles is necessary: How do the profiles look if no isothermal but a slowly increasing, continuous temperature ramp is applied? To what extent do the results depend on the scan rate?"

Response:

Following Reviewer's suggestion, we have now added a set of conventional thermogravimetry (TG) measurements on the model mTFF-SO₃-Na polymer at different ramp rates from 1 to 10°C min⁻¹ as suggested. The data show that the standard TG protocol cannot discriminate the first two water species at whatever scan rates, but the derivative spectrum clearly reveals two regimes. The results are presented in new Figures 2a and b. We have now improved our discussion of the segmented TG protocol. The reason for the segmentation is really to allow desorption of the lowest water fraction to finish before ramping to desorb the higher fractions. The fact that this works for all 20+ polymer systems tested suggests it would be generally useful. The isothermal plots at different temperatures are shown in Figure 4a. They provide the basis for our kinetics analysis.

2. "... more details on how the geometries have been determined and optimized and how many conformations have been considered needs to be given. Also the phrase "ground state geometry" is not clear to me."

Response:

This information is now provided in the Experimental. Our "ground state geometry" is the local energy minimum structure, not global minimum. In view of possible confusion, we have removed this term from our discussion, and used "local energy minimum" instead. Briefly, we seek to generate a set of 'metastable' structures to sample, perhaps somewhat haphazardly, a diversity of possible binding motifs as a guide to the interpretation of experimental activation energies and ν_{OH} vibrational frequencies. Briefly, in our methodology, the ion multiplet is generated and geometry-optimized to a local minimum, then water is added, randomly, to vicinity of the ion cluster, and the

geometry re-optimized, then water was removed, and the structure re-optimized. The energy difference between the two corresponding states is taken as the desorption energy of water. Several different seeds were used to generate several 'hydrated' states. We took the opportunity in the revision to add more computed structures to check for consistency of our methodology, see new Figure 6, Supplementary Figure 1, and Table 3. We are quite satisfied with the results.

3. "The spectra in Figure 8b&c are presumably difference spectra, it is however not fully clear how these have been obtained. In Figure 8c the experimental data are shown only over a limited wavenumber range. To judge the reliability of the background subtraction the data should be shown over the same range as in 8a. Only such the reader can judge the significance of the fit with Gaussians. For these fits also the fitting equation should be given. Just stating a Gaussian width of 60 cm^{-1} is ambiguous (e.g. square root of variance, or full width at half maximum)."

Response:

Yes, fixed. Data extending from below $3,000 \text{ cm}^{-1}$ to above $3,600 \text{ cm}^{-1}$ are displayed now in Fig 9c and 9d. You can see the CH vibration modes, which also serve as internal reference. The difference spectra in Fig 9b are simply $B - A$, the identities of A and B now clearly specified in the plots. Our Gaussian widths are the standard deviation widths. This is now explicitly stated in the equation in the text.

4. "On page 9 (lines 232-236) the authors argue that the absence of a cation effect in E_a rules out direct cation contact. This should be toned down, as it just suggests that the potential energy due to interaction with the cation is not the dominant contribution. Further, the author argue that this rules out the core hydration model. This seems not supported by the experiments, as the energy does not tell anything about the location of the water. The author also have not calculated the hydration energy in the core, but just of an ion-pair, thus the data to not provide any insights into that."

Response:

Thank you for this comment. We've reconsidered this aspect further. Two lines of evidence to abandon the core hydration model is now further clarified, with a caveat:

One, the computed binding energy of water to cation is large, of the order of a few tenths of an eV (0.7 eV for Na^+), and strongly dependent on cation size, but the experiment E_a is small, and shows little dependence on cation size, so the water is not directly coordinated to the cation, as required in the core hydration model.

Two, both the infrared and Raman spectra of the sorbed water indicate a large fraction of dangling OH bonds and H-bond configurations that are consistent with bonding to anion. This is also not compatible with the core hydration model.

Nevertheless, in the as-deposited state, some amount of core hydration occurs in the form of trapped water. We have now explained this better now.

The Referee is right that “the potential energy due to interaction with the cation is not the dominant contribution”. We have now analyzed this effect quantitatively. As shown in Figure 8, ion-dipole electrostatic binding energy can be quite large for small cations, decaying as $1/d^2$ to about 0.15 eV at 3.7 Å, the size of tetramethylammonium ion, but the electric field still perturbs the OH stretching frequencies by about 100 cm^{-1} (Fig 8). This can be observed in the infrared and Raman spectra (Fig 9c,d; Fig 10). We have clarified this proximal cation effect.

5. “I find discussion of the IR results on page 12 difficult to follow. What are the main differences between the three different spectra in Figure 8c and d. What types of water supposedly prevails for the samples in different conditions? What is the inferred binding state for type I and type II water? Only referring to “(inspection, Figure 7)” makes it extremely difficult to follow the arguments.”

Response:

Thank you for this comment. We have further improved the discussion. In brief, type-I water is hydrogen-bonded water clusters. Reversible type-II water is H-bonded primarily to anion at the ion cluster surface, under the influence of the cation; but irreversible type-II water, and type-III water, is dual-bonded to both anion and cation, or ‘caged’ between ion pairs, in the ion cluster.

6. “Also the sentence “The formation of extensive 3D H-bonded water clusters required in the core hydration model can also be definitively ruled out.” Is not clear. On what basis can it be ruled out? And why does the core hydration model require 3D H-bonded clusters at hydration levels as low as the once studied here?”

Response:

We thank the Referee for this comment. We have removed the statement. Core hydration model indeed produces 3D H-bonded clusters only at high hydration levels.

Reviewer 2

“The paper is well written and the analysis carefully done. Sufficient information is provided to enable other researchers to reproduce their results. Publication is recommended.”

Response:

We thank this Referee for the appreciation and kind comments...

REVIEWERS' COMMENTS

Reviewer #1 (Remarks to the Author):

The authors have addressed all my previous comments appropriately. I find the work very impressive and the findings are a key to understand the effect of water on the properties of polyelectrolytes. I therefore recommend publication after the authors have considered the following last minor remarks:

- On page 4: "Yet, the notion that water drives the formation of solvent-shared and then solvent-separated ion pairs from the onset of hydration does not accord with current understanding of ionic solutions"

Maybe the authors can be more specific on what aspects are inconsistent with our current understanding of electrolyte solutions?

- In the next sentence, the authors argue that electrolytes at 6M are thermodynamically unstable with a reference to ionic liquids. I am not sure if this can be generalized such. In particular, as far as I know, most ionic liquids are fully miscible with water – except those containing 'hydrophobic anions' such as PF₆⁻, NTf₂⁻ or larger fluorinated anions. Ionic liquids based on e.g. sulfate anions – which seem relevant to the present work – are reported to be fully miscible (see e.g. doi 10.1021/je060228n). Thus, the authors should phrase this paragraph more carefully.

- On page 9: the authors describe their fitting model as Arrhenius desorption, which I initially found confusing. Maybe a better phrase would be 'nth order kinetics with the rate constant obeying thermal activation based on Arrhenius' or something similar.

- In the caption of Fig 5 the authors note that their MD production runs were performed for 100ps at 100K. Could the authors provide some explanation why the temperature was such low, much lower than the experimental conditions?

Detailed Response to Referee Reports

The authors have addressed all my previous comments appropriately. I find the work very impressive and the findings are a key to understand the effect of water on the properties of polyelectrolytes. I therefore recommend publication after the authors have considered the following last minor remarks:

Reviewer 1

We thank Reviewer for kind remarks! His/her other suggestions and comments have now been fully adopted and addressed. Our line-by-line responses are as follows.

1. "... "Yet, the notion that water drives the formation of solvent-shared and then solvent-separated ion pairs from the onset of hydration does not accord with current understanding of ionic solutions"

Maybe the authors can be more specific on what aspects are inconsistent with our current understanding of electrolyte solutions?"

Response: Fixed. Thank you so much for your thoughtful comments that helped clarify our own thoughts. We have replaced the paragraph with one that better contrasts the hydration situation in ionic liquids with that in polyelectrolytes, focusing on the nature of the hydration, rather than our initial argument about thermodynamic instability. We were originally framing that argument based on the phase diagrams of related ionic liquids showing coexistence behaviour, but we agree that there exists also fully miscible systems that would invalidate this argument.

2. "... the authors argue that electrolytes at 6M are thermodynamically unstable with a reference to ionic liquids. I am not sure if this can be generalized such. In particular, as far as I know, most ionic liquids are fully miscible with water – except those containing 'hydrophobic anions' such as PF6-, NTf2- or larger fluorinated anions. Ionic liquids based on e.g. sulfate anions – which seem relevant to the present work – are reported to be fully miscible (see e.g. doi 10.1021/je060228n). Thus, the authors should phrase this paragraph more carefully."

Response: The Reviewer is right. We have redone the paragraph more carefully (see above), and with more complete literature references, and adjusted adjacent paragraphs accordingly.

3. "On page 9: the authors describe their fitting model as Arrhenius desorption, which I initially found confusing. Maybe a better phrase would be 'nth order kinetics with the rate constant obeying thermal activation based on Arrhenius' or something similar."

Response: The Reviewer is right. We have rephrased accordingly.

4. “In the caption of Fig 5 the authors note that their MD production runs were performed for 100ps at 100K. Could the authors provide some explanation why the temperature was such low, much lower than the experimental conditions?”

Response: Fixed. Originally, we used MM2, which gave many problems, including a fundamental flaw of not modelling ion-dipole interactions correctly. We have since migrated to OPLS4, which we found to be much better parametrized, and faster. We have employ this to re-run all the MD calculations, extending also to the R_FSIS system (new Figure 4). The runs are now conducted at 300 K over the nanosecond time scale.